# Prior Knowledge Guided Neural Architecture Generation

**Jingrong Xie** [1]   **Han Ji** [1]   **Yanan Sun**[⊠ 1]

## Abstract

Automated architecture design methods, especially neural architecture search, have attracted increasing attention. However, these methods naturally need to evaluate numerous candidate architectures during the search process, thus computationally extensive and time-consuming. In this paper, we propose a prior knowledge guided neural architecture generation method to generate high-performance architectures without any search and evaluation process. Specifically, in order to identify valuable prior knowledge for architecture generation, we first quantify the contribution of each component within an architecture to its overall performance. Subsequently, a diffusion model guided by prior knowledge is presented, which can easily generate high-performance architectures for different computation tasks. Extensive experiments on new search spaces demonstrate that our method achieves superior accuracy over state-of-the-art methods. For example, we only need 0.004 GPU Days to generate architecture with 76.1% top-1 accuracy on ImageNet and 97.56% on CIFAR-10. Furthermore, we can find competitive architecture for more unseen search spaces, such as TransNAS-Bench-101 and NATS-Bench, which demonstrates the broad applicability of the proposed method.

## 1. Introduction

Neural Architecture Search (NAS) is an effective approach to automatically designing neural architectures and has seen success in diverse tasks. NAS generally involves three main components (Elsken et al., 2019): search space defining possible architectures, search strategy exploring this vast space to identify promising candidates, and performance evalu-

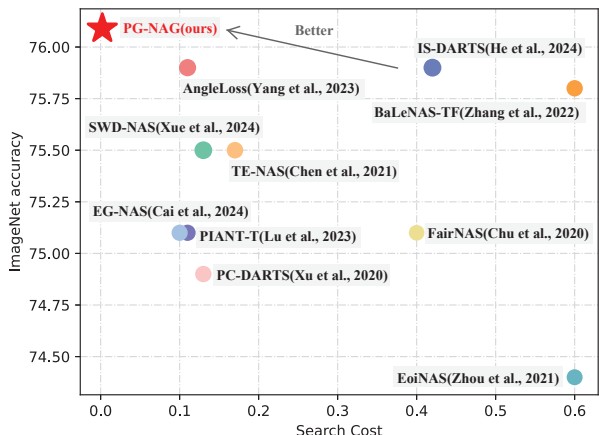

*Figure 1.* Search cost and performance comparison of our proposed PG-NAG method against the state-of-the-art NAS methods on ImageNet.

ation assessing each architecture to find the optimal one. However, the exhaustive search for an optimal architecture is computationally demanding due to the vast search space, which often contains over $10^{10}$ possible architectures (Cai et al., 2019). Furthermore, evaluating each candidate architecture requires training from scratch on a dataset to obtain accuracy, leading to significant time and resource costs.

To alleviate these costs, several methods have been introduced to accelerate architecture evaluation, such as performance predictors (Ji et al., 2024) and zero-shot methods (Peng et al., 2024). These methods aim to predict the performance of architectures without training from scratch. However, even with faster evaluations, current NAS methods still need to evaluate architectures individually, which remains time-intensive, especially with a large number of candidates. Therefore, minimizing computational costs remains a central challenge in NAS research.

This leads to growing interest in the Neural Architecture Generation (NAG) paradigm, which reduces costs by avoiding the exhaustive search approach and instead focuses on directly generating high-performance architectures. For instance, AutoBuild (Mills et al., 2024) trains a predictor to rank the performance of the modules in architecture, thus

[1]Department of Computer Science Sichuan University, Chengdu, China. Correspondence to: Yanan Sun <ysun@scu.edu.cn>.

*Proceedings of the 42nd International Conference on Machine Learning*, Vancouver, Canada. PMLR 267, 2025. Copyright 2025 by the author(s).

constructing architecture by combining high-valued modules in each layer. DiffusionNAG (An et al., 2024) trains a diffusion model to generate architectures and utilizes a predictor to guide the generative direction to high-performance architectures, thus achieving significant acceleration and improving search performance.

However, these NAG methods still rely on predictors to guide the generation, as they lack the methods to investigate the search space toward optimal solutions systematically. Without such predictors, these models would randomly select architecture components, resulting in inefficient exploration. Furthermore, training these predictors will introduce additional computational overhead.

This paper aims at generating promising architectures without relying on additional predictors. Inspired by (Xiao et al., 2022), we try to utilize prior knowledge to analyze the importance of operations to improve the efficiency of NAG. We propose a novel framework to generate architectures with prior knowledge guidance rather than additional predictors, entitled PG-NAG. A key consideration for PG-NAG is ensuring the quality of prior knowledge. This quality contains two aspects: one is that prior knowledge should be derived from reliable sources, and the other is that it should accurately reflect the contributions of various components within the architecture. PG-NAG not only provides relevant and precise guidance but also allows us to effectively generate architectures across various search spaces without additional training.

As shown in Figure 1, PG-NAG can generate promising architectures and significantly reduce computational complexity. To achieve this, our method learns the key components of high-performance architectures from benchmark datasets to form prior knowledge, which then serves as a guiding condition to construct the diffusion model. In the meantime, architectures from benchmark datasets eliminate the cost of training architectures to obtain accuracy. Notably, PG-NAG achieves this efficiency with minimal computational resources, requiring only 0.004 GPU days to generate well-performing architectures.

- We propose a novel method for neural architecture generation, i.e., PG-NAG, relying on learning component knowledge from high-performance architectures sampled from benchmark datasets. Significantly different from traditional automated design approaches, PG-NAG generates high-performance architectures without iterative search and additional training costs.

- We design a new diffusion model to generate architectures, which can leverage prior knowledge to guide the generation of architectures. This helps generate architectures that meet the needs of different tasks.

- Experimental results demonstrate that PG-NAG successfully generates architectures across multiple unseen search spaces, achieving superior performance compared to state-of-the-art NAS and NAG methods. The generated architectures not only achieve 97.56% accuracy on CIFAR-10, but also demonstrate 76.1% top-1 accuracy when transferred to ImageNet.

## 2. Related Works

### 2.1. Neural Architecture Search (NAS)

NAS can discover the optimal architecture from a set of possible architectures automatically and can be mainly classified into reinforcement learning-based NAS, evolutionary-based NAS, and gradient-based NAS (Elsken et al., 2019). However, regardless of which NAS algorithm is used, many architectures require being evaluated, resulting in high evaluation costs. To alleviate this, various accelerating methods have been proposed, such as performance predictor (Lu et al., 2023) and zero-shot (Peng et al., 2024). In any case, they still need to evaluate massive architectures iteratively, PG-NAG bypasses this limitation and offers a more efficient way to generate architectures automatically.

### 2.2. Diffusion Models

Diffusion models are probabilistic generative models that progressively destroy data by injecting noise, then learn to reverse this process for sample generation (Yang et al., 2023a). Since the initial diffusion model is designed to learn the data distribution at noise level, it is often modified for adapting to various controlled generation scenarios. Previous studies have already made preliminary attempts to control generation. For example, Classifier-free guidance methods (Ho & Salimans, 2022) enhance the quality and diversity of generated samples by mixing score estimates from conditional and unconditional diffusion models. Furthermore, the research (Vignac et al., 2022) has shown that noise distributions closely resembling the sample data distribution can lead to improved performance. Building on these findings, PG-NAG employs prior knowledge to modify the noise distribution, thereby better guiding the diffusion model in generating high-performance architectures.

## 3. Methods

In this section, we present PG-NAG, a method designed to generate high-performance architectures for various tasks by leveraging a small selection of pre-existing high-performance architectures from benchmark datasets. We first present the overview. Then, we show how to obtain prior knowledge and the generation progress based on the prior knowledge, as illustrated in Figure 2.

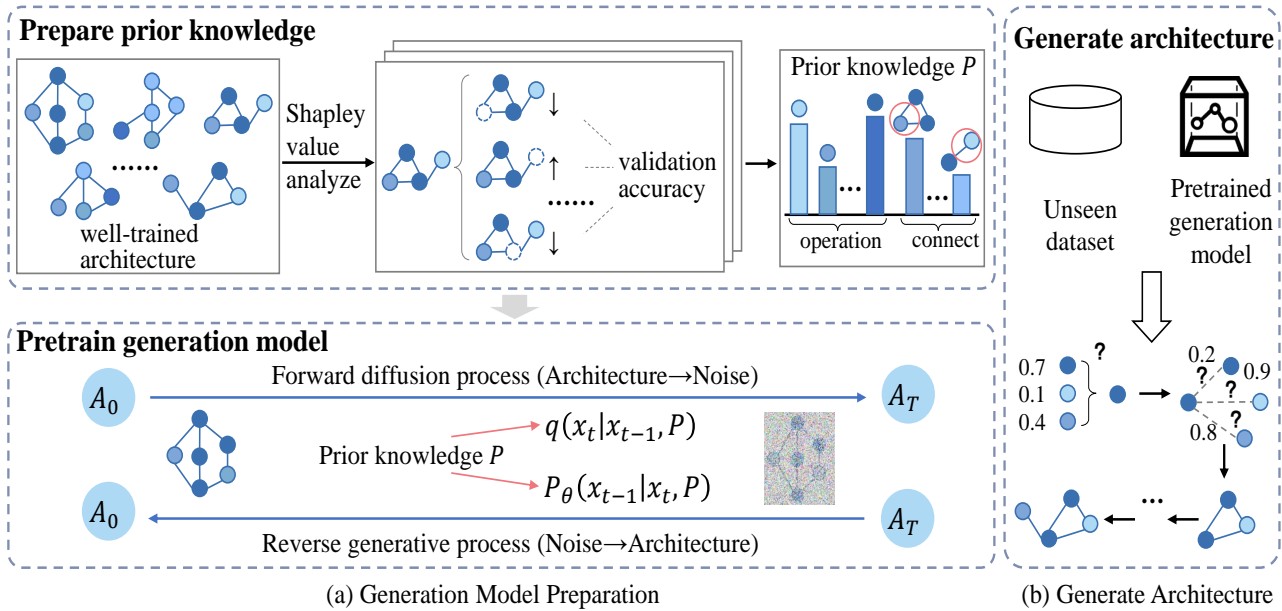

*Figure 2.* An illustration of PG-NAG, which contains the preparation and utilization of architecture generation model, where the preparation includes the preparation of prior knowledge and pretrain of the model.

---

**Algorithm 1** Overall Framework of PG-NAG

**Input**: High-performance architectures $\mathcal{A}$ from benchmarks, parameter $\theta$ for diffusion model, the number of time steps T

**Output**: The resulting generated high-performance architecture $A^*$

1: Let architectures $A$ represent as graphs, each can be represented as $\mathcal{GR} = \langle \mathcal{O}, \mathcal{C} \rangle$.
2: Obtain prior knowledge $\mathcal{K}$ = Shapley value ($\mathcal{GR} = \langle \mathcal{O}, \mathcal{C} \rangle$) as Equation (3) and (4)
3: Sample $t \sim \text{Uniform}(1, \ldots, T)$
4: Sample $\epsilon \sim \mathcal{N}(0, I)$
5: Take gradient step on $A_t \sim p_\theta(A_{t-1}|A_t, \mathcal{K}, \epsilon)$ as Equation (5)
6: Sample $A_T \sim p_\theta(A_T) = \mathcal{N}(0, I)$
7: **for** $t \leftarrow T$ to 1 **do**
8:  Compute $\mu_t(A_T, A, t)$ from $\epsilon(A_T, A, t)$
9:  Sample $A_{t-1} \sim q_\theta(A_t|A_{t-1}, \mathcal{K}, \mu_t)$ as Equation (6)
10: **end for**
11: **return** $A^* = A_0$

---

### 3.1. Overview

In PG-NAG, to avoid the computational cost of evaluating pre-existing architectures, we utilize a subset of architectures $\mathcal{A}$ from NAS benchmarks, which contains many architectures and corresponding accuracy values. We first obtain prior knowledge $\mathcal{K}$ from these architectures using Shapley values, which quantify the contribution of each operation

and connection in architectures. PG-NAG then proposes a diffusion model to newly generate high-performance architectures under the guidance of prior knowledge $\mathcal{K}$, ensuring that the generation process is both informed and efficient, without further evaluations. As illustrated in Algorithm 1, each architecture $A \in \mathcal{A}$ is represented as a graph $\mathcal{GR} = \langle \mathcal{O}, \mathcal{C} \rangle$, $\mathcal{O} = \{o_1, \ldots, o_n\}$ is a set of nodes, $\mathcal{C} = \{c_{ij} | (i, j) \subseteq |\mathcal{O}| \times |\mathcal{O}|\}$ is a set of edges, representing operations and connections between operations, respectively.

Subsequently, we provide a explanation of the proposed diffusion model. Considering this task in the Denoising Diffusion Probabilistic Model (DDPM) (Ho et al., 2020), a small Gaussian noise is gradually injected into data as a Markov chain. This process ultimately converges into a white noise distribution after $T$ iterations, leading to the following forward diffusion process:

$$q(A_{1:T}|A_0, \mathcal{K}) = \prod_{t=1}^{T} q(A_t|A_{t-1}, \mathcal{K}) \tag{1}$$

where $A_0, \ldots, A_T$ denote a sequence of latent variables with the same dimension, $t$ is the index of diffusion steps, as the data $A_0 \sim q(A_0)$.

In the reverse generative process, the noise is gradually removed from the data to recover architecture $A$ from the white noise. This is achieved by iteratively applying a carefully designed reverse process that targets the original data

distribution. Specifically, at each step $t$, we aim to approximate the conditional distribution $p_\theta(A_{t-1}|A_t, \mathcal{K})$, which represents the probability of the architecture before noise has been added. This distribution is calculated under the condition of the current noisy state and the prior knowledge $\mathcal{K}$. Thus, this process is essentially a data generation process as a reverse dynamics of the above diffusion process. The reverse process with a neural network parameterized by $\theta$ is represented by Equation (2).

$$p_\theta(A_{0:T-1}, A_T) = \prod_{t=1}^{T} p_\theta(A_{t-1}|A_t, \mathcal{K}) \qquad (2)$$

During this reverse process, the start of generating architectures $p_\theta(A_T)$ is set as a standard Gaussian and then is used as the prior distribution, i.e., from which new samples are generated. For an architecture, it can be generated by drawing chaotic statuses $A_T$ from $p_\theta(A_T)$, and then iteratively refined through the reverse Markov kernels $p_\theta\{A_{0:T-1}, A_T\}$.

## 3.2. Prior Knowledge Learning

To effectively guide the generation process, PG-NAG leverages prior knowledge obtained from high-performance architectures in benchmark datasets, eliminating the need for additional training. The process begins by sampling top-$k$ architectures from popular benchmarks to study the components in high-performance architectures. With the Shapley value evaluation, we quantify the contribution of each operation and connection to overall architecture performance.

Due to the interdependence between operations and connections within an architecture, the Shapley values reflect these interactions, treating the architecture as a cooperative game. Specifically, we evaluate the contributions of two components: operations and the connections between operations. The connections are represented as edges. In architecture, edges are directional to represent the flow of information, and learning the interactions of operations is critical to obtain important information contained in the architecture (Chen et al., 2021b). While connections and operations significantly impact architecture performance, their individual contributions alone do not guarantee high-performance architectures. Instead, we focus on learning one-hop subgraphs for each node to effectively capture the connections and their architectural relevance. The Shapley value measures the average marginal contribution of operations and one-hop subgraphs to the overall performance of an architecture. It is calculated by evaluating the performance disparity between the complete architecture and those without specific operations or subgraphs.

Specifically, each architecture $A$ is represented as a graph, which contains operations $o^i \in \mathcal{O}$ and one-hop subgraphs $g^j \in \mathcal{G}$. The Shapley value is utilized to distribute the total performance gains $V(N)$ to each operation and one-hop subgraphs in $N$, while $N$ represents the number of operations and subgraphs in this architecture. Therefore, a set of individual operations $N_\mathcal{O}$ and a set of individual subgraphs $N_\mathcal{G}$ can be modeled as players in the cooperative game, where all players work together towards the architecture performance $V(N)$, with $N = N_\mathcal{O} + N_\mathcal{G}$. A value function $V$ maps each subset of players $S \subseteq N$ to a real value $V(S)$ which represents the expected payoff that the players can obtain by cooperation. The Shapley value $\phi_o^i(P)$ and $\phi_g^j(P)$ for an operation $o^i$ and a one-hop subgraph $g^j$ can be computed as follows, respectively:

$$\phi_o^i(P) = \frac{1}{|N_\mathcal{O}|} \sum_{S \subseteq N_{o^i}} \frac{V(S \cup \{o^i\}) - V(S)}{\binom{|N_\mathcal{O}|-1}{|S|}} \qquad (3)$$

$$\phi_g^j(P) = \frac{1}{|N_\mathcal{G}|} \sum_{S \subseteq N_{g^j}} \frac{V(S \cup \{g^j\}) - V(S)}{\binom{|N_\mathcal{G}|-1}{|S|}} \qquad (4)$$

After the validation accuracy is used as the value function $V$ to measure the architecture performance, the performance of the whole architecture is the sum of contributions of operations and one-hop subgraphs, i.e., $\sum_{o^i \in N_\mathcal{O}} \phi_o^i + \sum_{g^j \in N_\mathcal{G}} \phi_g^j = V(N)$. If the operation does not influence performance, whether added or removed, its contribution is considered zero and we define $\phi_o^i = 0$. For instance, the zeroize operation in NAS-Bench-201 and zero operation in DARTS have no effect on the final performance. Furthermore, if two different operations or one-hop subgraphs have the same impact on the performance of architecture, they are assigned with equal contributions. Based on these properties, the Shapley value can quantify individual contributions uniquely.

Based on the above calculation, we can quantify the specific contribution of operations and connections to the performance of architectures. This allows us to pinpoint the operations and connections that are important in the generation of high-performance architectures. With a clear understanding of these critical elements, we are better equipped to generate high-performance architectures.

## 3.3. Diffusion Model with Prior Knowledge

To generate high-performance architectures, we design a conditional diffusion model that incorporates prior knowledge for guidance. The model operates as a latent variable model with two processes: the forward diffusion process and the reverse generative process. In the forward process, Gaussian noise gradually injects the input architecture into a white noise distribution over $T$ iterations. The reverse process then removes this noise step-by-step, recovering the original architecture. By leveraging prior knowledge, the

model can approximate the architecture distribution more effectively than a pure Gaussian prior. During the reverse generative process of the diffusion model, the prior knowledge obtained from the Shapley value evaluation can guide generated architectures that are closer to the original architecture.

We first formalize the prior knowledge to clearly demonstrate the guiding role of prior knowledge in the conditional diffusion model. The prior knowledge is formalized as a set of data-dependent priors $\mathcal{K} = \left\{ O_{1:N}^i; \mathcal{G}_{1:N}^{ij}, \sum_{1:N} \right\}$, which includes operations and connections that have important contribution to high performance, $N$ denotes the number of prior knowledge components. $O_i$ is prior operations and $\mathcal{G}^i$ is the one-hop subgraph around operation $O_i$ in high performance architectures, and $\sum_{1:N}$ is the prior covariance matrix.

Next, we will describe how the diffusion process and generative process will be adjusted under the guidance of prior knowledge. Our diffusion model aims to build the distribution $p_\theta(A) = p_\theta(A, \mathcal{K})$, as the process of operations $O$ and one-hop subgraph $\mathcal{G}$ are similar, we only focus the derivation on the diffusion and generative process of operations $O$. For connections, we use the same method to get the generated ones.

We follow the definition and notations related to the noise schedule $\alpha_t, \beta_t, \hat{\alpha}_t, \hat{\beta}_t$ in (Ho & Salimans, 2022). With prior knowledge as guidance, the forward and reverse process can be defined as follows:

$$q(A_t|A_{t-1},\mathcal{K}) = \prod_{i=1}^{N_A} \prod_{n=1}^{N} \mathcal{N}\left( A_{t,n}; A_{t-1,n}, \beta_t \sum_n \right) \quad (5)$$

$$p_\theta(A_{t-1}|A_t,\mathcal{K}) = \prod_{i=1}^{N_A} \prod_{n=1}^{N} \mathcal{N}\left( A_{t-1,n}; \right.$$
$$\left. \tilde{\mu}_t\left(A_{t,n}, A_{0,n}|\mathcal{K}\right), \tilde{\beta}_t \sum_n \right) \quad (6)$$

where $\tilde{\mu}_t\left(A_{t,n}, A_{0,n}\right) = \frac{\sqrt{\alpha_t}(1-\bar{\alpha}_{t-1})}{1-\bar{\alpha}_t} A_{t,n} + \frac{\sqrt{\bar{\alpha}_{t-1}}\beta_t}{1-\bar{\alpha}_t} A_{0,n}$.

The prior distribution in the forward diffusion process can be expressed as:

$$p(A_T|\mathcal{K}) = \prod_{i=1}^{N_A} \prod_{n=1}^{N} \mathcal{N}\left( A_T; \mu_n, \sum_n \right) \quad (7)$$

As illustrated in the above equations, the proposed diffusion model learns the knowledge of architecture design based on prior knowledge. To enable the learned prior knowledge to be effectively utilized by the diffusion model, we construct a graph convolutional network and multilayer perceptron (MLP) to extract features of operations and one-hop

graphs, respectively. After completing the forward process of the diffusion model, the diffusion model can probabilistically select key operations and one-hop graphs around the selected operations during the reverse process. In this way, the proposed diffusion model can generate efficient architectures optimized for specific task requirements.

## 4. Experiments

In this section, we evaluate PG-NAG on five new and different search spaces, including DARTS (Liu et al., 2018), NATS-Bench (Dong et al., 2021), TransNAS-Bench-101 (Duan et al., 2021), NAS-Bench-ASR (Mehrotra et al., 2021), and NAS-Bench-NLP (Klyuchnikov et al., 2022). We first explain the experimental settings in Section 4.1, then compare PG-NAG with state-of-the-art algorithms in Section 4.2, and finally analyze the generated architectures and perform an ablation study in Sections 4.3 and 4.4, respectively. More experimental results on other search spaces can be found in **Appendix C.1** and **Appendix C.2**.

### 4.1. Experimental Settings

To generate high-performance architectures for new search spaces and unseen tasks, we first learn prior knowledge from three popular benchmarks. Specifically, we select top-20 high-performance architectures from each of the three popular benchmarks, which are NAS-Bench-101 (Ying et al., 2019), NAS-Bench-201 (Dong & Yang, 2020b), and NAS-Bench-301 (Zela et al., 2020). Please note that NAS-Bench-101 and NAS-Bench-201 contain architecture with accuracy, and NAS-Bench-301 contains accuracies of architectures in DARTS search space by using a proxy model to evaluate architectures without training. Therefore, we can learn the contribution of components to high-performance architectures across different search spaces without additional training for architectures. To implement the PG-NAG method, we integrate a graph convolution network with 64 hidden layers to extract operation features in our diffusion model. Additionally, a three-layer MLP is utilized to capture knowledge of connections in architectures. To validate the effectiveness of PG-NAG in five new search spaces, we design five architectures for each search space. Subsequently, depending on the convention of the different search spaces, we select the highest value or the average value of these five architectures as the validation result. For different search spaces, we used different methods to compare with PG-NAG, including different search methods in NAS and efficient evaluation methods in NAS. More details about experimental implementation are in **Appendix B**.

### 4.2. Experimental Results of Search Spaces

**Results on DARTS.** Table 1 presents the results of architectures generated by PG-NAG within the extensive DARTS

Table 1. Performance comparison with state-of-the-art architectures on ImageNet and CIFAR-10 in DARTS search space.

| METHODS | COST (GPU-DAY) | TEST ERROR (%) | | PARAMS (M) | FLOPs (M) |
|---|---|---|---|---|---|
| | | IMAGENET | CIFAR-10 | | |
| DARTS (LIU ET AL., 2018) | 4 | 73.3 / 91.3 | 97.24±0.09 | **4.7** | 574 |
| SNAS (XIE ET AL., 2018) | 1.5 | 72.7 / 90.8 | 97.15±0.02 | 4.3 | 474 |
| P-DARTS (CHEN ET AL., 2019) | 0.3 | 75.6 / 92.6 | 97.50 | 4.9 | 557 |
| GDAS (DONG & YANG, 2019) | 0.21 | 74.0 / 91.5 | 97.18 | 5.3 | 545 |
| PC-DARTS (XU ET AL., 2020) | 0.13 | 74.9 / 92.2 | 97.43±0.07 | 5.3 | 586 |
| ISTA-NAS (YANG ET AL., 2020) | 4.2 | 76.0 / 92.9 | 97.46±0.05 | 5.65 | 638 |
| SDARTS-ADV (CHEN & HSIEH, 2020) | 1.3 | 74.8 / 92.2 | 97.52±0.02 | 3.3 | - |
| FAIR DARTS (CHU ET AL., 2020B) | 0.4 | 75.1 / 92.5 | 97.46 | 4.8 | **541** |
| DARTS+PT (WANG ET AL., 2021) | 0.8 | 74.5 / 92.0 | 97.52 | 4.6 | - |
| TE-NAS (CHEN ET AL., 2021A) | 0.17 | 75.5 / 92.5 | 97.37±0.06 | 5.4 | - |
| EOINAS (ZHOU ET AL., 2021) | 0.6 | 74.4 / 91.7 | 97.50±0.10 | 5.0 | - |
| BALENAS-TF (ZHANG ET AL., 2022) | 0.6 | 75.8 / 92.7 | 97.5±0.07 | 5.3 | 597 |
| PRE-NAS (PENG ET AL., 2022) | 0.6 | 76.0 / 92.6 | 97.51±0.09 | 6.2 | - |
| EAEPSO (YUAN ET AL., 2023) | 4 | 73.1 / - | 97.49 | 4.9 | - |
| MOEA-PS (XUE ET AL., 2023) | 2.6 | 73.6 / 91.5 | 97.23 | 4.7 | - |
| ANGLELOSS (YANG ET AL., 2023B) | 0.11 | 75.9 / **92.9** | 97.44 | 5.9 | - |
| PIANT-T (LU ET AL., 2023) | 0.11 | 75.1 / 92.5 | 97.46±0.08 | 5.2 | 583 |
| SWD-NAS (XUE ET AL., 2024) | 0.13 | 75.5 / 92.4 | 97.49 | 6.3 | - |
| EG-NAS (CAI ET AL., 2024) | 0.1 | 75.1 / - | 97.47 | 5.2 | - |
| IS-DARTS (HE ET AL., 2024A) | 0.42 | 75.9 / 92.9 | 97.44±0.04 | 6.4 | - |
| PG-NAG (AVERAGE) | **0.004** | - / - | 97.48 ± 0.07 | 5.5 | 554 |
| PG-NAG (BEST) | **0.004** | **76.1** / 92.7 | **97.56** | 5.5 | 554 |

search space, which is evaluated on ImageNet (Deng et al., 2009) and CIFAR-10 (Krizhevsky et al., 2009). All results for SOTA methods in Table 1 are sourced directly from their respective papers, where only the best accuracy (denoted as xxx) or the mean accuracy (denoted as xxx±xx) is reported. PG-NAG achieves state-of-the-art performance, with the highest ImageNet accuracy and CIFAR-10 accuracy. Remarkably, the architecture generated by PG-NAG only needs 0.004 GPU days, or approximately 5.76 minutes. In terms of model complexity, architectures generated by PG-NAG maintain a competitive parameter size and FLOPs, striking a balance between performance and resource efficiency. These results highlight the effectiveness of PG-NAG in generating high-quality architectures with minimal computational resources. Further visualizations and architecture stability analysis can be found in **Appendix C.3** and **Appendix C.4**, respectively.

Table 2. Performance comparison with state-of-the-art architectures in NATS-Bench search space.

| METHODS | COST (S) | CIFAR-10 | CIFAR-100 | IMAGENET16-120 |
|---|---|---|---|---|
| ENAS | 13,315 | 93.76±0.00 | 70.67±0.62 | 41.44±0.00 |
| BOHB | 12,000 | 93.94±0.28 | **72.00**±0.86 | **45.70**±0.86 |
| RSPS | 7,587 | 91.05±0.66 | 68.26±0.96 | 40.69±0.36 |
| DARTS | 29,902 | 65.38±7.84 | 60.49±4.95 | 36.79±7.59 |
| GDAS | 28,926 | 93.23±0.58 | 68.17±2.50 | 39.40±0.00 |
| SETN | 31,010 | 92.72±0.73 | 69.36±1.72 | 39.51±0.33 |
| PG-NAG | **4,147** | **93.91**±0.08 | 70.96±0.14 | 44.73±0.35 |

**Results on NATS-Bench.** We evaluate PG-NAG within the NATS-Bench search space $sss$ to assess its efficiency compared to several state-of-the-art NAS methods benchmarked on NATS-Bench. These comparison methods employ a range of different search strategies, all aiming at identifying high-performance architectures. As shown in Table 2, PG-NAG achieves competitive performance across three datasets—CIFAR-10, CIFAR-100, and ImageNet-16-120 (Chrabaszcz et al., 2017) in NAS-Bench-201 search space, demonstrating superior results without a search process. In addition, the cost of PG-NAG maintains a significantly lower search cost of 4,171 seconds compared to other methods. The results demonstrate that PG-NAG can generate architectures that achieve competitive accuracies, highlighting its efficiency and effectiveness in neural architecture generation.

**Results on TransNAS-Bench-101.** In order to verify the effectiveness and generability of the architecture generated by PG-NAG on different visual tasks, we validate it on the TransNAS-Bench-101 search space. We compare PG-NAG with four NAS search methods (Duan et al., 2021) and one NAS acceleration method (He et al., 2024b). As shown in Table 3, PG-NAG shows strong performance in Class Object and Room Layout. In order to use a metric to measure the combined performance of the above methods on the seven tasks, we add an average rank column to the right of Table 3. Specifically, we compute the ranking of architectures for each task. These rankings are then averaged to evaluate the

*Table 3.* Performance comparison with state-of-the-art architectures in TransNAS-Bench-101 search space. Room layout's L2 loss is multiplied by a factor of 100 for better readability. The rightmost column reports the average percentile rank of searched networks (Avg.Rank) in the benchmark, averaged across all target tasks.

| TASK | CLASS OBJECT | CLASS SCENE | AUTO-ENCODING | SURFACE NORMAL | SEMANTIC SEGMENT | ROOM LAYOUT | JIASAW | AVERAGE RANK |
|---|---|---|---|---|---|---|---|---|
| METRIC | ACC.↑ | ACC.↑ | SSIM↑ | SSIM↑ | MIOU↑ | L2LOSS↓ | ACC.↑ | |
| DT | 42.03 | 49.80 | 51.20 | 55.03 | 22.45 | 66.98 | 88.95 | 42.53 |
| RS | 45.16 | 54.41 | 55.94 | 56.85 | 25.21 | 61.48 | 94.47 | 13.70 |
| PPO-TFS | 45.19 | 54.37 | 55.83 | 56.90 | 25.24 | 61.38 | 94.46 | 13.53 |
| PPO-TRANSFER | 44.81 | 54.15 | 55.70 | 56.60 | 24.89 | 62.01 | - | 14.26 |
| ROBOT | 45.59 | 54.87 | 55.42 | 57.44 | 26.27 | 61.16 | 94.82 | 16.43 |
| PG-NAG | 46.32 | 54.38 | 54.12 | 57.30 | 25.03 | 60.09 | 94.40 | **12.23** |

generalization and effectiveness of architectures searched or generated by different methods. Although other methods perform well on specific tasks, PG-NAG shows competitive performance across the seven tasks, demonstrating its excellent generalization capabilities across different tasks.

*Table 4.* Comparation of test validation analysis on NAS-Bench-ASR and NAS-Bench-NLP.

| METHODS | TEST PER (%) | | | TEST LOG PERPLEXITY | | |
|---|---|---|---|---|---|---|
| | MIN. | MAX. | AVG. | MIN. | MAX. | AVG. |
| NASBOT | 21.89 | 22.02 | 21.93±3.81 | 4.58 | 4.67 | 4.61±0.03 |
| NPENAS | 21.88 | 21.91 | 21.89±0.96 | 4.55 | 4.67 | 4.58±0.03 |
| BANANAS | 21.86 | 21.99 | 21.89±3.79 | 4.6 | 4.67 | 4.63±0.02 |
| REA | 21.90 | 22.25 | 21.95±0.10 | 4.62 | 4.67 | 4.63±0.02 |
| PG-NAG | 21.58 | 21.86 | 21.78±0.13 | 4.53 | 4.66 | 4.57±0.02 |

**Results on NAS-Bench-ASR.** In order to validate the performance of PG-NAG on the unseen task of speech recognition, we compare it with four different NAS methods on the NAS-Bench-ASR search space. The architecture generated by PG-NAG will be validated on the TIMIT speech recognition dataset (Garofolo et al., 1993), using Phoneme Error Rate (PER) as the performance metric. As shown in Table 4, architectures generated by PG-NAG have lower error rates than other methods, consistently outperforming them in terms of minimum, maximum, and average values. Therefore, PG-NAG can generate architectures with more stable results and lower speech recognition error rates.

**Results on NAS-Bench-NLP.** To evaluate the effectiveness of the architectures generated by PG-NAG on the unseen language modeling task, we conduct a comparison with four other NAS methods within the NAS-Bench-NLP search space. The architectures are tested on the Penn Tree Bank dataset, with test log perplexity serving as the performance metric. As illustrated in Table 4, PG-NAG achieves the lowest test log perplexity. The variance in the performance of architectures generated PG-NAG is also the lowest compared to other methods, further suggesting that it generates stable and reliable architectures.

*Table 5.* Statistics of the generated architectures. Each method generates 1, 000 architectures.

| DATASET | STATES. | RANDOM | DIFFUSIONNAG | PG-NAG |
|---|---|---|---|---|
| | MAX | **94.37** | **94.37** | 94.36 |
| CIFAR-10 | MEAN | 87.12 | 94.13 | **94.15** |
| | MIN | 10.00 | 86.44 | **87.08** |
| | MAX | 72.74 | **73.51** | **73.51** |
| CIFAR-100 | MEAN | 61.59 | 70.34 | **70.79** |
| | MIN | 1.00 | 58.09 | **66.08** |

## 4.3. Distribution of generated architectures

We further explore the distribution of the accuracy of architectures generated by PG-NAG. Firstly, we compare the distribution of 1, 000 generated architectures with 1, 000 randomly selected architectures in NAS-Bench-201, as shown in Figure 3. Then, we compare the stability of the generated architectures with random architecture sampling and DiffusionNAG (An et al., 2024) in Table 5. The results show that the distribution of the generated architectures has low variance, indicating a concentrated distribution characteristic with a tendency for clustering in the high-performance part. Meanwhile, both the mean and the minimum of the accuracy of the generated architectures are higher than that of DiffusionNAG. This means that PG-NAG can target the generation of high-performance architectures when dealing with specific tasks. To make the comparison clearer, we delete some low-performance architectures obtained by random selection, the original figures are in **Appendix C.6**.

## 4.4. Ablation Study

**Components of PG-NAG.** We evaluate the impact of three core components of PG-NAG: prior knowledge guidance, operation feature extraction method, and connection feature extraction method. The experimental results on CIFAR-10, CIFAR-100, and ImageNet16-120 in NAS-Bench-201 are shown in Table 6. The prior knowledge denotes the

*Table 6.* Abalation study about components of PG-NAG on NAS-Bench-201.

| PRIOR KNOWLEDGE | OPERATION FEATURE | CONNECTION FEATURE | CIFAR-10 | | CIFAR-100 | | IMAGENET16-120 | |
|---|---|---|---|---|---|---|---|---|
| | | | VALID | TEST | VALID | TEST | VALID | TEST |
| ✗ | GCN | ✓ | 90.45 | 93.31 | 71.17 | 71.13 | 45.63 | 45.77 |
| ✓ | GIN | ✓ | 91.50 | 94.37 | 73.31 | 73.09 | 45.59 | 46.33 |
| ✓ | GAT | ✓ | 91.34 | 94.00 | 72.98 | 72.48 | 45.23 | 46.20 |
| ✓ | GCN | ✗ | 91.35 | 94.30 | 72.77 | 72.30 | 45.53 | 46.44 |
| ✓ | GCN | ✓ | 91.55 | 94.36 | 73.49 | 73.51 | 46.37 | 46.34 |

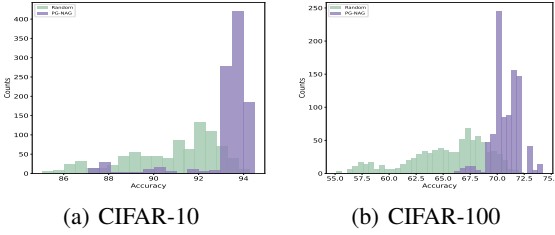

(a) CIFAR-10  (b) CIFAR-100

*Figure 3.* The distribution of generated architectures.

generation of architectures without the guidance of prior knowledge. The operation feature involves replacing the original graph convolution network for operation extraction with a graph isomorphic network or a graph attention network. The connection feature represents the learning method of connections used by MLP for learning one-hop subgraphs is replaced with a new MLP that learns for connections only. These results highlight that prior knowledge guidance leads to the greatest performance drop, indicating its critical role in PG-NAG.

*Table 7.* Comparison with different diffusion models on NAS-Bench-201.

| METHODS | NOISE SCHEDULE | NUMBER OF TIMESTEP | CIFAR-10 | CIFAR-100 | IMAGENET |
|---|---|---|---|---|---|
| DDPM | LINEAR | 1,000 | 93.28 | 70.23 | 41.88 |
| PG-NAG | SIGMOID | 500 | 92.73 | 69.38 | 42.83 |
| | SIGMOID | 800 | 94.22 | 73.17 | 46.48 |
| | LINEAR | 1,000 | 93.42 | 70.90 | 45.33 |
| | SIGMOID | 1,000 | 94.36 | 73.51 | 46.34 |

**Diffusion model.** To evaluate the effectiveness of the diffusion model in PG-NAG, we conduct a comparative analysis with the standard DDPM diffusion model. Additionally, we examine the impact of key components in our diffusion model, i.e., the noise schedule and the number of time steps. As shown in Table 7, PG-NAG significantly outperforms DDPM on CIFAR-10, CIFAR-100, and ImageNet16-120. For the noise schedule, PG-NAG utilizes a sigmoid schedule, which we compare to the linear schedule used in DDPM. Additionally, we test different time step settings, including

500, 800, and 1,000 steps. While larger time steps generally improve the accuracy of the sampling process, a larger time step usually results in higher computational cost. Therefore, in order to ensure the efficiency of architecture generation, we strike a balance between computational efficiency and the performance of architectures. These findings confirm that the diffusion model designed in PG-NAG is highly effective and well-suited for architecture generation tasks.

*Table 8.* Compare the quality and quantity prior knowledge on NAS-Bench-201.

| METHOD | CIFAR-10 | | CIFAR-100 | | IMAGENET16-120 | |
|---|---|---|---|---|---|---|
| | VALID | TEST | VALID | TEST | VALID | TEST |
| $k$=10 | 89.68 | 92.31 | 68.84 | 69.49 | 43.22 | 42.60 |
| $k$=20 | 91.55 | 94.36 | 73.49 | 73.51 | 46.37 | 46.34 |
| $k$=30 | 91.53 | 94.22 | 73.13 | 73.17 | 46.32 | 46.68 |
| $k$=50 | 91.61 | 94.37 | 72.75 | 73.22 | 45.56 | 46.71 |
| RANDOM 20 | 82.42 | 87.11 | 57.06 | 56.95 | 30.70 | 30.57 |

**Prior knowledge.** To verify the significance of prior knowledge, we conduct ablation experiments on the quality and quantity of the selected architectures in NAS-Bench-201. Firstly, we evaluate the impact of varying the number of high-performance architectures $k$ used as prior knowledge to guide the generation model. We evaluate PG-NAG with $k = 10, 20, 30$, and 50 architectures. As shown in Table 8, the results indicate that $k = 50$ yields the highest valid and test accuracy in CIFAR-10 and test accuracy in ImagNet-16-120. These results suggest that $k = 20$ is a practical and effective choice, as it achieves near-optimal performance across datasets. Increasing $k$ to 30 or 50 provides an additional benefit, causing unnecessary computational cost when using larger prior knowledge sets. Instead, we select a lower but effective number to learn prior knowledge.

Secondly, to evaluate the quality of selected architectures, we select 20 random architectures instead of high-performance architectures. We find that the performance of architectures declines compared with using high-performance architectures as prior knowledge. These comparisons demonstrate that the quality of prior knowledge is crucial for guiding the architecture generation pro-

cess. High-performance architectures provide meaningful insights that enable PG-NAG to achieve superior results, whereas random architectures fail to provide such guidance.

## 5. Conclusion

In this paper, we propose a method to automatically generate high-performance architectures guided by prior knowledge. It leverages an analysis of high-performance architectures to formulate prior knowledge. Then the knowledge serves as a guiding principle for the diffusion model, enabling it to automatically generate architectures optimized for various tasks. Extensive experimental results on new search spaces show that PG-NAG generation process is highly efficient on unseen tasks without costly and time-consuming training. In addition, the effectiveness of the components in PG-NAG is verified by the ablation study.

## Impact Statement

This paper presents work whose goal is to advance the field of Machine Learning. There are many potential societal consequences of our work, none which we feel must be specifically highlighted here.

## Acknowledgments

This work was supported by National Natural Science Foundation of China under Grant 62276175 and Innovative Research Group Program of Natural Science Foundation of Sichuan Province under Grant 2024NSFTD0035.

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

# A. Details of PG-NAG.

## A.1. Architectures Encoding for Different Search Spaces

Since we have chosen architectures from three different search spaces for learning prior knowledge, we first need a unified representation of the architectures. To unify the architectures in different search spaces, we design a generalizable encoding for different search spaces following the convention. Specifically, we encode all operations from different search spaces into vectors using the one-hot encoding method. For operations in different search spaces but with similar functionality, we use the same encoding. For instance, we encode *max_pooling* in NAS-Bench-101 and *avg_pooling* in NAS-Bench-201 with the same encoding. Additionally, NAS-Bench-101 has operations on nodes while NAS-Bench-201 and DARTS have operations on edges. Therefore, we transfer operations on nodes to standardize. This encoding approach allows us to encode architectures from three search spaces in a uniform standard.

## A.2. Dependence on Prior Knowledge

The success of PG-NAG relies on the quality of the prior knowledge. As we all know, different operations in architecture are not equally important (Chen et al., 2021b). Therefore, some methods have tried to analyze the operations in architecture, such as (Wang et al., 2021; Xiao et al., 2022). Furthermore, by analyzing the effect of different operations in DARTS, DARTS-(Chu et al., 2020a) discovers that skip connections have a clear advantage over other candidate operations and treats it as prior knowledge. Therefore, this method adds a decaying auxiliary skip connection. Similarly, iDARTS (Zhang et al., 2021) analyzes the preferences of different operations as prior knowledge. Based on prior knowledge, iDARTS interposes a static BN layer between the node input and the operations to maintain the balance between different operations. Therefore, prior knowledge has been used in the process of NAS to search for optimal architectures.

In this paper, we recognize that PG-NAG relies on carefully selected architectures to obtain prior knowledge. This knowledge is derived from three search spaces of different sizes, while the architectures in these search spaces have been validated on three different datasets, which provides us with a broad reference for generating new architectures. By learning from these diverse architectures, we obtain representative prior knowledge, which enables our method to not only perform well in these three search spaces but also to achieve good results in other unseen search spaces. This shows that the prior knowledge we learned based on these architectures is highly generalizable and flexible.

# B. Experimental Details

## B.1. Search Spaces

In this section, we describe the search spaces that are used to verify the effectiveness of PG-NAG.

*DARTS* (Liu et al., 2018) search space is one of the most popular search space. It contains a total of $10^{18}$ unique architectures. These architectures are always evaluated on three datasets: CIFAR-10, CIFAR-100, and ImageNet.

*NAS-Bench-201* (Dong & Yang, 2020a) is a widely used benchmark that includes a total of $15,625$ unique architectures. These architectures are trained and evaluated on CIFAR-10, CIFAR-100, and ImageNet16-120 datasets.

*TransNAS-Bench-101* (Duan et al., 2021)is a relatively recent addition to NAS benchmarks. It evaluates architectures across seven vision tasks: Scene Classification, Object Classification, Autoencoding, Surface Normal, Semantic Segmentation, Room Layout, and Jigsaw Puzzle. A total of $51,464$ architectures are evaluated across these tasks with comprehensive training details.

*NAS-Bench-101* (Ying et al., 2019) is the first public architecture benchmark for NAS research. It consists of $423,624$ unique architectures, all of which are trained and fully evaluated multiple times on the CIFAR-10 dataset.

*NAS-Bench-ASR* (Mehrotra et al., 2021) is a NAS benchmark for Automatic Speech Recognition (ASR). It contains $8,242$ unique models trained on the TIMIT audio dataset for three different epochs, with each model starting from three different initializations.

*NAS-Bench-NLP* (Klyuchnikov et al., 2022) is a NAS benchmark focused on the language modeling task in natural language processing. It includes $14,000$ RNN-like architectures, each trained and validated on the Penn Tree Bank (PTB) dataset.

### B.2. Datasets

In this section, we describe the validation datasets that are used to verify the effectiveness of PG-NAG.

For computer vision validation datasets, we utilize three widely used datasets: CIFAR-10, CIFAR-100, and ImageNet, which are all image classification datasets. The number of images in CIFAR-10 and CIFAR-100 is $60,000$. CIFAR-10 has ten classes, each containing $6,000$ images. CIFAR-100 has 100 classes, each containing 600 images. ImageNet is a massive dataset comprising over $12,197,122$ images. ImageNet has 1000 classes, each contains images between 732 and $1,300$. ImageNet-16-120 downsamples ImageNet to $16\times16$ pixels, from which selects all images with label $\in [1, 120]$ to construct ImageNet-16-120. ImageNet-16-120 contains $151,700$ training images, $3,000$ validation images, and $3,000$ test images with 120 classes.

For the automatic speech recognition task, we valid architectures on TIMIT dataset. It consists of recordings of 630 speakers of 8 dialects of American English each reading 10 phonetically rich sentences. It also comes with the word and phone-level transcriptions of the speech.

For the natural language processing task, we valid architectures on PTB dataset, which is one of the most known and used corpus for the evaluation of models for sequence labelling. It consists of annotating each word with its Part-of-Speech tag. It has $38,219$ setences and $912,344$ tokens for training, $5,527$ sentences and $131,768$ tokens for validation, and $5,462$ sentences and $129,654$ tokens for testing.

### B.3. Implementation Details

In the search process in NAS-Bench101, NAS-Bench-201, and TransNAS-Bench-101, we generate a cell and stack it into an architecture according to the benchmark rules. In the DARTS search space, we generate a cell and set the number of initial convolutional channels to 36. We optimize the architecture weights using stochastic gradient descent with an initial learning rate of $0.025$ and a single Consine annealing learning rate schedule. All experiments were done on Linux Ubuntu 18.04, using Nvidia 3090 GPUs.

## C. Additional Experimental Results

### C.1. Results on NAS-Bench-201

*Table 9.* Performance comparison with state-of-the-art architectures on CIFAR-10, CIFAR-100, and ImageNet16-120 in NAS-Bench-201 search space.

| METHODS | CIFAR-10 | | CIFAR-100 | | IMAGENET16-120 | |
|---|---|---|---|---|---|---|
| | VALID | TEST | VALID | TEST | VALID | TEST |
| DARTS | 39.77 | 54.30 | 38.57 | 38.97 | 18.87 | 18.41 |
| SGNAS | 90.18 | 93.53 | 70.28 | 70.31 | 44.65 | 44.98 |
| PC-DARTS | 89.96 | 93.41 | 67.12 | 67.48 | 40.83 | 41.31 |
| iDARTS | 89.96 | 93.58 | 70.57 | 70.83 | 40.38 | 40.89 |
| DrNAS | 91.55 | 94.36 | 73.49 | 73.51 | 46.37 | 46.34 |
| FairNAS | 90.97 | 93.23 | 70.94 | 71.00 | 41.09 | 42.19 |
| BANANAS | - | 94.37 | - | 73.51 | - | - |
| AngelLoss | 90.19 | 93.16 | 71.70 | 70.48 | 41.93 | 43.04 |
| EG-NAS | 90.17 | 93.58 | 70.90 | 70.98 | 45.18 | 46.59 |
| L2NAS | 91.47 | 94.28 | 73.02 | 73.09 | 46.58 | 47.03 |
| DiffusionNAG | - | **94.37** | - | **73.51** | - | - |
| PG-NAG | **91.55** | 94.36 | **73.49** | 73.51 | **46.37** | **46.34** |

We applied PG-NAG to NAS-Bench-201 to automatically generate high-performance architecture. We compare PG-NAG against various popular NAS algorithms and one NAG method: DARTS and its variants (Liu et al., 2018; Xu et al., 2020; Zhang et al., 2021; Chu et al., 2021), SGNAS (Huang & Chu, 2021), DrNAS (Chen et al., 2020), BANANAS (White et al., 2021), AngelLoss (Yang et al., 2023b), EG-NAS (Cai et al., 2024), L2NAS (Mills et al., 2021), and DiffutionNAG (An et al., 2024). As shown in Table 9, all searches or generates are performed on CIFAR-10, and the output architecture is then trained and evaluated on each of CIFAR-10, CIFAR-100, and ImageNet16-120. These comprehensive experimental results

indicate that our method surpasses the other baselines.

## C.2. Results on NAS-Bench-101

*Table 10.* Comparison with other NAS methods on CIFAR-10 using the NAS-Bench-101 search space.

| METHODS | RANKING (%) | TEST ACC (%) |
|---------|-------------|--------------|
| REA | 0.384 | 93.64 |
| ANGELLOSS | 0.242 | 93.70 |
| E2EPP | 0.132 | 93.77 |
| HAAP | 0.004 | 94.09 |
| RENAS | 0.011 | 94.01 |
| SEMINAS | 0.011 | 94.01 |
| NPENAS | 0.002 | 94.14 |
| AG-NAS | 0.002 | 94.18 |
| BRP-NAS | 0.001 | 94.22 |
| BANANAS | 0.004 | 94.08 |
| GMAE-NAS | 0.002 | 94.14 |
| BONAS | 0.002 | 94.14 |
| PG-NAG | **0.001** | **94.23** |

We evaluate the architecture generated by PG-NAG on NAS-Bench-101. We compare PG-NAG with 12 NAS methods. As outlined in Table 10, the generated architecture demonstrates a performance improvement over previous methods such as REA (Real et al., 2019), AngelLoss (Yang et al., 2023b), and E2EPP (Sun et al., 2019). To set up a basic benchmark, we also compare PG-NAG with HAAP (Liu et al., 2021), ReNAS (Xu et al., 2021), SemiNAS (Tang et al., 2020), NPENAS (Wei et al., 2022), AG-NAS (Lukasik et al., 2022), BRP-NAS (Dudziak et al., 2020), BANANAS (White et al., 2021), GMAE-NAS (Jing et al., 2022), and BONAS (Shi et al., 2020). Moreover, the proposed method PG-NAG achieved the highest accuracy architecture $94.23\%$ in this search space.

## C.3. Architecture Generated for DARTS and NAS-Bench-201 Search Space

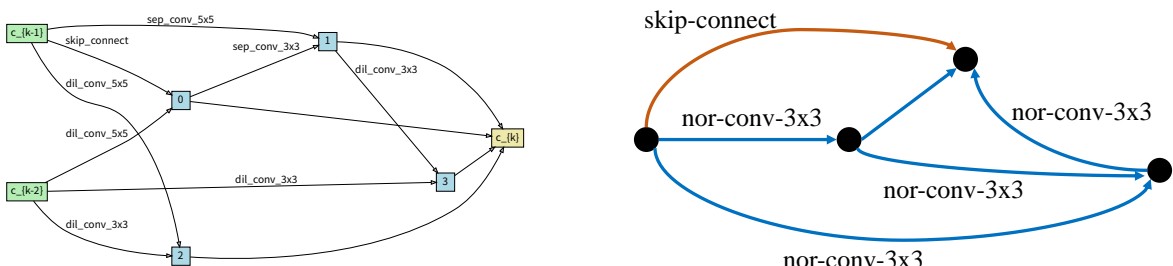

*Figure 4.* Cell architecture found by PG-NAG on CIFAR-10 and ImageNet datasets.

*Figure 5.* Architecture found by PG-NAG for NAS-Bench-201 search space.

We use our PG-NAG to generate high-performance architecture in the DARTS search space based on the CIFAR-10 dataset at first. Then we transfer this architecture to the ImageNet dataset in the DARTS search space. Figures 4 demonstrate the neural architectures generated by PG-NAG for the CIFAR-10 and ImageNet datasets. The generated architectures contains three $dil\_conv\_3 \times 3$ and two $dil\_conv_5 \times 5$, and only one $skip\_connect$. This is consistent with the fact that the high-performance architectures searched by existing NAS methods contain a large number of convolutional operations.

Similarly, Figures 5 demonstrates the neural architectures generated by PG-NAG for the CIFAR-10, CIFAR-100, and ImageNet datasets. The generated architecture consists of four $nor\_conv\_3 \times 3$ and one $skip\_connect$, which closely resembles the high-performing architectures observed in NAS-Bench-201. Specifically, these architectures often include multiple $3 \times 3$ convolutions, highlighting the potential for similar performance patterns in the generated design.

## C.4. Architecture Stability Analysis

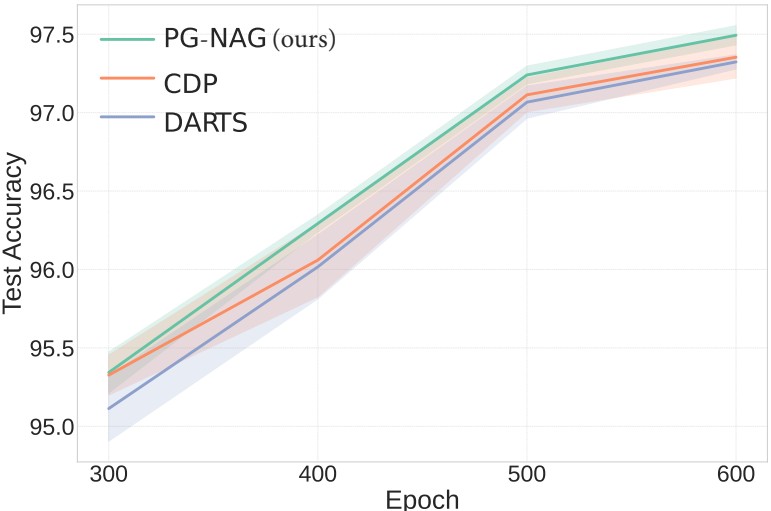

*Figure 6.* Comparison results on existing methods on CIFAR-10 on DARTS search spaces.

To verify the stability of the architectures generated by PG-NAG, we designed five different architectures for the CIFAR-10 dataset in the DARTS search space. We run these architectures five times and calculate their performance variance. In addition, we utilize the architectures generated by DARTS and CDP method (Liu et al., 2022), which performs well in the DARTS search space. All experiments are run five times independently to ensure the reliability of the results. As shown in Figure 6, it can be seen that the PG-NAG achieves the best performance and demonstrates remarkable stability.

## C.5. Visualization for Architecture Accuracy Drop

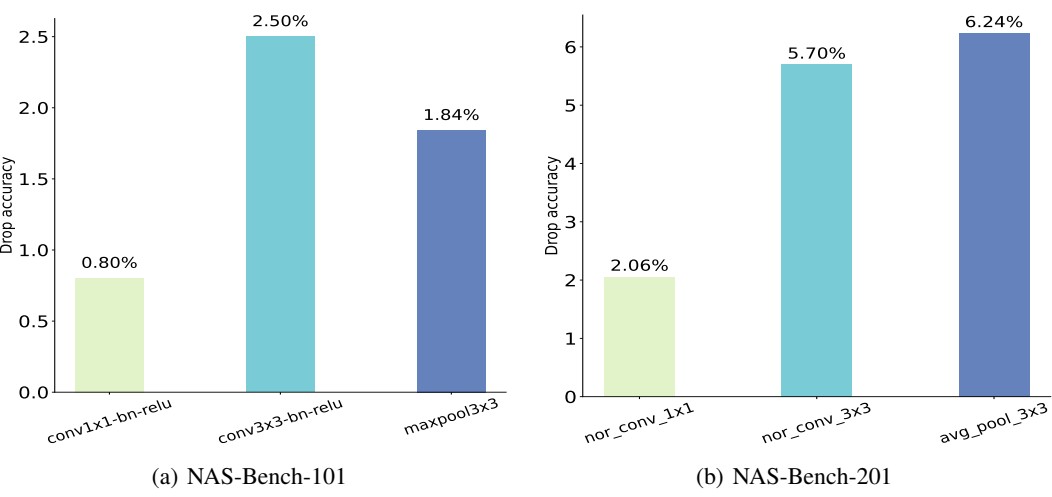

*Figure 7.* Visualization of operation influence on architetcure performance in different search space.

To better understand how different operations affect the performance of architectures, which can help explain the selection of operations in our generated architectures. We remove operations in NAS-Bench-101 and NAS-Bench-201 to observe the impact of different operations on the performance of the architecture. Specifically, for each architecture, we compute performance by removing one operation at a time, progressively eliminating each operation individually (e.g., first removing the first operation, then the second, and so on).

As shown in Figure 7(a) for NAS-Bench-101, the *Conv_3x3* operation has the most significant impact on performance, underscoring its importance in achieving high accuracy. A similar trend is observed in NAS-Bench-201 in Figure 7(b), where the $3 \times 3$ convolution operation enhances model performance. These findings help us make informed decisions in selecting the most effective operations for generating high-performing architectures.

### C.6. Architecture Distribution of Generated Architectures

To demonstrate the effectiveness of PG-NAG in generating high-performance architectures stably, we conducted an experiment where $1,000$ architectures were generated using PG-NAG within the NAS-Bench-201 search space. Additionally, we randomly selected $1,000$ architectures from the same search space for comparison. Then we evaluate these architecutes on CIFAR-10 and CIFAR-100 datasets. As shown in the Figure 8, the two figures are a more complete distribution map than the Figure 3 in the main text Section 4.3. Architectures generated by PG-NAG consistently exhibit high accuracy, indicating that PG-NAG is capable of generating a large number of architectures with competitive performance stably. In contrast, the randomly selected architectures demonstrate a broader range of performance, with some architectures performing significantly worse. This demonstrates the ability of PG-NAG to stably generate high-performance architectures.

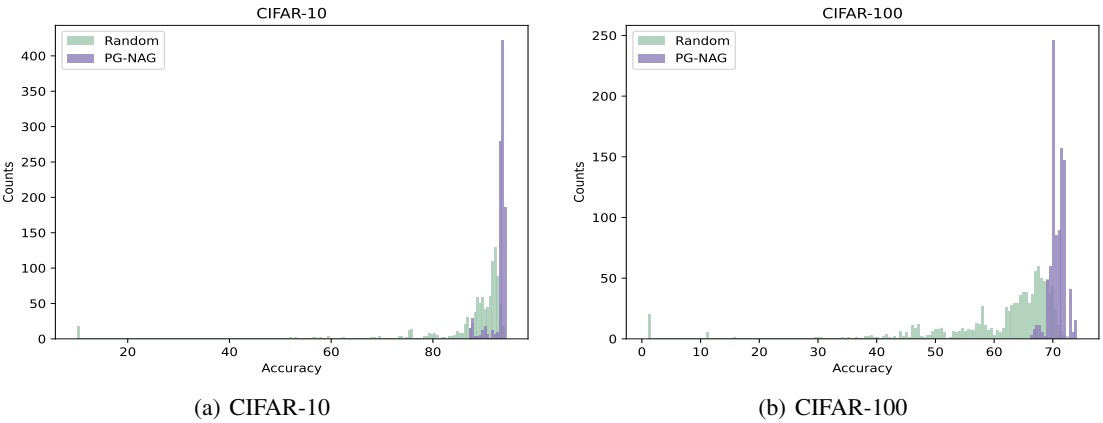

(a) CIFAR-10          (b) CIFAR-100

*Figure 8.* The distribution of generated architectures on CIFAR-10 and CIFAR-100 in NAS-Bench-201.

