# OpenReview forum: "Prior Knowledge Guided Neural Architecture Generation"
_ICML.cc/2025/Conference — ICML 2025 poster_

### Official Review · Reviewer_xYiC · 2025-02-19

**Overall Recommendation:** 2

**Summary:**

The authors propose a Neural Architecture Generation (NAG) method based on Prior knowledGe (PG, for PG-NAG). NAG techniques extend Neural Architecture Search (NAS) to discover the features such as operations and subgraphs that contribute to high performance. This is achieved using Shapley values. Specifically, importance's are initialized as the prior knowledge of the diffusion model and the Shapley values from the top-20 architectures are used to initalize prior knowledge. PG-NAG is evaluated on several benchmarks such as NAS-Bench-101, NATS-Bench, TransNAS-Bench, etc.

**Post-Rebuttal Response**
I thank the authors for their detailed response. While my score will remain as weak reject, I quote and emphasize the direct language of how ICML perceives such a score: "Weak reject (i.e., leaning towards reject, but could also be accepted)"

*Reasons to raise score*
- The rebuttal is pretty well-detailed and provides good factual clarification on the workings of the paper.
- The justification for the performance drop of PG-NAG compared to other techniques is mostly that PG-NAG is faster/more efficient and the authors emphasize this. They are encouraged to place further emphasis on this advantage in the paper, e.g., BOHB taking 12k seconds per Table 2.

*Reasons to lower score*
- The rebuttal's key weakness is the response regarding Table 1. To quote the rebuttal "We apologize for the confusion. We report the best performance, selecting the top architecture from five runs of PG-NAG." That the authors are showing the best result for **their** method yet the average result for **other** methods is bizarre and conspicuous. As the main result for the paper, it is literally an apples-to-oranges comparison that cannot be overlooked.

**Claims And Evidence:**

A claim of this paper is that search spaces are large, necessity smarter techniques for performance evaluation. While it is indeed true that large search spaces prohibit evaluation, there are existing techniques that overcome this burden.

**Essential References Not Discussed:**

For NAS-Bench-101 comparison in Table 10, a missing related work would be GA-NAS [3] which also obtains 94.23%.
For the idea of finding relevant architecture subcomponents you have NAS-Bowl [4] which is not cited.

**Experimental Designs Or Analyses:**

Experimental design is sound but not noteworthy.
A primary weakness of this paper is that its results are not even incremental. The method does not provide clear and convincing proof of its efficacy to merit acceptance. Some examples:
- Table 1: Line 260-263 (right column) states "To validate the effectiveness of PG-NAG in five new search spaces, we design five architectures for each search space", yet Table 1 only features one result from PG-NAG for ImageNet or CIFAR-10. Is it a mean value? A max?
- Table 2 method loses on CIFAR-100 ad ImageNet16-120.
- Table 3 the method loses on every task except Class Object and Room Layout
- Table 5 the method loses compared to Random and DiffusionNAG in terms of max performance.

**Methods And Evaluation Criteria:**

The method is trained and evaluated on several standard benchmarks like DARTS, etc. The evaluation experimental setup raises no alarms.

**Other Comments Or Suggestions:**

References:

[1] "DiffusionNAG: Predictor-guided Neural Architecture Generation with Diffusion Models" - ICLR'24

[2] "Building Optimal Neural Architectures using Interpretable Knowledge" - CVPR'24

[3] "Generative Adversarial Neural Architecture Search" - IJCAI-21

[4] "Interpretable Neural Architecture Search via Bayesian Optimisation with Weisfeiler-Lehman Kernels" - ICLR'21

**Other Strengths And Weaknesses:**

Presentation quality of this paper is definitively below the bar for an A* conference like ICML. As stated before the method is hard to understand and grasp in key sections, e.g., computation of Shapley values and `visualizing` how the method actually works.

Additionally, float formatting leaves much to be desired. Fig. 1/2 and Tab. 1 in the main manuscript are decently formatted, but the rest are not, e.g., either too small scalebox, too small text.

**Questions For Authors:**

What factors influence architecture selection criteria exactly?

**Relation To Broader Scientific Literature:**

PG-NAG extends the idea of directly generating or building high-performance architectures from a search space as opposed to using a search algorithm. There are several related works in this field such as DiffusionNAG [1], a primary related work that is cites. There is also AutoBuild [2], which is cited but not compared to (different benchmarks). This paper also lightly deals with the concept of finding the good/bad architecture subcomponents, thus falls under the banner of interpretable NAS, like AutoBuild and others.

**Theoretical Claims:**

This is difficult to evaluate as the paper's methodology is not well-written and difficult to understand, even after performing multiple passes over the methodology section. Specifically, better visual examples would help a lot.

One idea the reviewer takes issues with is that this method relies on knowing which architectures are good ahead of time, and which are bad. PG-NAG trains using only a small subset of cherry-picked architectures.

---

> ### Author Rebuttal · Authors · 2025-03-31
>
> We sincerely thank you for the recognition of our effective evaluation. We are also grateful for the valuable feedback.
> ## Theoretical Claims
> >Visual examples
>
> We add a Flowchart (_https://anonymous.4open.science/r/PGNAG/flowchart.png_).
> >Concerns about prior knowledge
>
> PG-NAG aims to train high-performance architectures from benchmarks and transfer the knowledge to generate architectures in other datasets. Experiments show that if we select architectures from a narrow performance range, the generated architectures still achieve competitive results (weakness 1 in Reviewer PwDK). This demonstrates that PG-NAG can learn the design principles.
>
> ## Experimental Designs
> >Table 1
>
> We apologize for the confusion. We report the best performance, selecting the top architecture from five runs of PG-NAG.
> >Table 2
>
> PG-NAG only lose to BOHB on CIFAR-100 and ImageNet16-120. However, it is important to note that PG-NAG runs $2.89$x faster than BOHB, i.e., BOHB requires about $12,000$ seconds yet PG-NAG only takes $4,147$ seconds to generate the high-performance architecture.
> >Table 3
>
> Though PG-NAG does not outperform other methods on every task, it achieves the highest average rank among all methods. Meanwhile, PG-NAG also shows significant gains on Class Object and Room Layout. The results demonstrate the generalization capability of it across diverse tasks.
> >Table 5
>
> Although PG-NAG shows a marginal 0.01% lower max performance compared to Random and DiffusionNAG, it is important to note that DiffusionNAG requires querying an architecture on the target dataset, while PG-NAG does not. Thereby enhancing the generalizability of PG-NAG. Compared to random, PG-NAG has higher minimum and mean values, and can consistently generate high-performance architectures.
> ## Supplementary Material
> It's important to note that DiffusionNAG and L2NAS require querying additional architectures on the target dataset, while PG-NAG does not. Specifically, DiffusionNAG queries one architecture, and L2NAS queries $1,000$ architectures. The extra querying demands additional computational resources and reduces the generalization capabilities. To compare, we utilize the same number of queried architectures with the two methods in PG-NAG. The accuracies are 94.31% for NAS-Bench-101, 94.37%, and 47.31% for CIFAR-10 and ImageNet-16-120 in NAS-Bench-201, which are higher than that of DiffusionNAG and L2NAS.
> ## Essential References
> We added the two methods in the new manuscript. GA-NAS has the same accuracy as PG-NAG, while NAS-Bowl underperforms at 94.2% accuracy compared to PG-NAG. However, GA-NAS requires an additional 150 architectures in NAS-Bench-101 compared to PG-NAG.
> ## Weakness
> >Presentation quality
>
> Thank you and we will thoroughly polish the manuscript. Subsequently, we provide a detailed explanation of the computation process for Shapley values and how the PG-NAG works.
> * Shapley values: Shapley values are used to quantify the importance of each operation to overall performance. We provide a detailed flowchart of this computation process in Figure (_https://anonymous.4open.science/r/PGNAG/shapley%20value.png_). Specifically, we remove a specific operation and measure the performance difference compared to the original architecture. This performance drop reflects the marginal contribution of the operation, the view of the performance drop is visualized in Appendix Figure 7. After computing Shapley values, we further visualize the relationships among these operations through heatmaps for NAS-Bench-101 (_https://anonymous.4open.science/r/PGNAG/Shapley%20values/NAS-Bench-101.pdf_), NAS-Bench-201 (_https://anonymous.4open.science/r/PGNAG/Shapley%20values/NAS-Bench-201.pdf_), and NAS-Bench-301(_https://anonymous.4open.science/r/PGNAG/Shapley%20values/NAS-Bench-301.pdf_).
>
> * Visualization of PG-NAG: We add a flowchart (https://anonymous.4open.science/r/PGNAG/flowchart.png). It begins with constraints of search space. For example, architectures are constrained to have six operations and four nodes in NAS-Bench-NLP. Then a diffusion model is trained for architecture generation. This diffusion model begins with extracting prior knowledge from the three benchmarks using Shapley values, then incorporates this guidance into the model to learn the architecture design principles.
>
> >Format of the manuscript
>
> We modified all the charts in the manuscript.
> ## Question:
> The influencing factors are as follows:
> * Prior knowledge: The quality of prior knowledge influences the architecture design principles learned by the PG-NAG (weakness 1 of Reviewer PwDK).
> * Noise schedule: Different noise schedules in the diffusion model affect the generation of architectures (weakness 2 of Reviewer PwDK).
> * Learning methods for operations and connections: We utilize GCN to learn the features of operations in the diffusion model. Cause the features of connections are hard to learn, we learn the features of subgraphs instead. The ablation study is in Table 6 in the main text.

---

### Official Review · Reviewer_VjZy · 2025-03-09

**Overall Recommendation:** 3

**Summary:**

This paper presents a novel method to enhance neural architecture generation using diffusion models. Instead of relying on predictor-based approaches, the authors train a diffusion model on graph representations of high-performing architectures. They further integrate explicit prior meta-knowledge extracted through Shapley value analysis to quantify the contribution of each component. This guidance steers the model to focus exclusively on generating high-quality architectures. Experimental results demonstrate that the method achieves promising performance improvements.

**Claims And Evidence:**

Most of the paper’s claims are supported by extensive empirical evidence. However, a couple of points warrant further scrutiny:

- As noted in Lines 96-99, the authors train their generative model solely on high-performing architectures. This approach could limit the diversity of the derived prior knowledge. A deeper discussion or additional experiments examining the potential limitations and diversity issues of this focus would strengthen the claim.
- Although the authors leverage prior knowledge to guide architecture generation with experimental results supporting this approach the evidence (see Table 8 in Section 4.4) suggests that the performance gains may primarily stem from extracting knowledge from high-performing architectures. This raises the question of whether the observed improvements are due to the superior quality of the training data rather than the guidance mechanism itself.

**Essential References Not Discussed:**

The authors provide a comprehensive review of the most relevant work in this field, effectively contextualizing their contributions. This paper

**Experimental Designs Or Analyses:**

Overall, the experimental framework is robust and well-aligned with established practices in NAS research. The authors evaluate their method on popular NAS datasets and benchmarks, comparing it with several existing approaches. However, one concern is that DiffusionNAG which also employs a diffusion-based approach but relies on an accuracy predictor is not included in Table 1 and only appears in later results. Including DiffusionNAG in the initial comparison would provide a clearer and more comprehensive evaluation of the proposed method relative to similar diffusion-based techniques.

**Methods And Evaluation Criteria:**

The proposed methods and evaluation criteria are well-suited to address the challenge of efficient neural architecture generation.

**Methodology**

 Unlike existing diffusion-based architecture generation methods that rely on accuracy predictors derived from graph representations, the authors introduce a novel approach that integrates explicit prior knowledge. This prior knowledge is extracted from high-performing architectures using Shapley value analysis, which quantifies the contribution of each neural network operation. By combining this information with the graph representation of architectures, the diffusion model is trained to focus exclusively on generating high-quality designs.

**Evaluation Criteria**

The authors evaluate their method across several well-established NAS benchmarks, including DARTS, NAS-Bench-201, TransNAS-Bench-101, NAS-Bench-ASR, and NAS-Bench-NLP. They use standard metrics such as top-1 accuracy, computational cost (measured in GPU days), parameter counts, and FLOPs. This comprehensive evaluation framework provides a robust comparison against state-of-the-art approaches and demonstrates the efficiency and effectiveness of the proposed method.

**Other Comments Or Suggestions:**

- Conduct a thorough investigation of the diversity of the generated architectures relative to the pretrained architectures.
- Include DiffusionNAG in Table 1 for a direct performance comparison.
- Explore the potential benefits of incorporating dataset-specific knowledge into the generation process.

**Other Strengths And Weaknesses:**

**Strengths:**
- **Efficiency in Generation:** By incorporating prior knowledge, the method reduces the need for iterative evaluation or reliance on accuracy predictors during the architecture generation process.
- **High-Quality Outputs:** The generated architectures achieve competitive performance, demonstrating that the guidance from high-performing designs is effective.

**Weaknesses:**

- **Dependency on Prior Knowledge Quality:** The method’s success heavily relies on a highly curated set of high-performing architectures. If the source data is biased or unrepresentative, the performance could be adversely affected.
- **Limited Architectural Diversity:** Training solely on high-performing architectures may constrain the model’s ability to generate diverse and innovative designs, potentially limiting the exploration of novel architectures. This lack of diversity is shown in Figure 3 and Table 5. since the method focused only on high performing architecture.
- **Lack of Dataset Conditioning:** Unlike DiffusionNAG, which uses dataset-conditioned sampling to tailor generation for unseen datasets, this method operates via blind generation, potentially limiting its adaptability to new tasks.
- **Attribution of Performance Gains:** There is a concern that the observed performance improvements might primarily stem from the high-quality training data rather than the inherent advantages of the proposed method itself.

**Questions For Authors:**

1. **Architecture Retrieval:**
   The sampling process appears to resemble a retrieval of high-performing architectures, since the conditioning is fixed based on a set of already high-performing designs. Could you clarify how this method differs from a simple retrieval mechanism, and what ensures that truly novel architectures are generated?

2. **Variance and Diversity:**
   How do you explain the observed low variance in the generated architectures? Could this be an indication that the method is overly focused on high-performing designs, thereby potentially limiting diversity and innovation?

3. **Comparison with DiffusionNAG:**
   DiffusionNAG uses a dataset-conditioned accuracy predictor, while your approach removes this predictor. How does your method ensure that it generates high-performing architectures for unseen datasets? Is it possible that the performance gains are primarily due to the fact that the training architectures perform similarly well on unseen datasets?

4. **Incorporating Dataset Information:**
   What are the challenges in integrating dataset-specific information into your generation process? It seems that blindly generating architectures may be effective only if the pretrained set is highly representative. Could you discuss the potential difficulties or limitations in incorporating dataset conditioning into your approach?

**Relation To Broader Scientific Literature:**

By integrating prior knowledge into the generation process, the authors extend and enrich the current scientific literature in neural architecture design. Specifically, they replace a task-conditioned accuracy predictor with a prior derived from high-performing architectures. However, this approach raises the question of whether training exclusively on high-performing architectures is sufficient to overcome the benefits offered by dataset-conditioned accuracy predictor-based models. While the contribution may appear marginal at first glance, the underlying idea holds promise and could pave the way for further improvements.

**Theoretical Claims:**

The proposed method builds upon conventional diffusion models, as seen in previous architecture generation approaches, and its claims are consistent with established principles in the conditional diffusion literature.

---

> ### Author Rebuttal · Authors · 2025-03-31
>
> Thank you for recognizing our efficiency, high-quality outputs, and robust experiments. We are grateful for the constructive feedback.
> ## Claims and Evidence
> >Potential limitations and diverse generation
>
> We discuss potential limitations regarding the quality of prior knowledge (weakness 1 in Reviewer PwDk). Then we prove PG-NAG can generate diverse architecture (weakness 3 in Reviewer PwDk).
> >Effectiveness of guidance mechanism
>
> We conduct experiments that indicate the guidance mechanism indeed plays a critical role. Specifically, we replace prior knowledge with architectures of 40%-50% and 80%-90% accuracy. PG-NAG generates architectures whose performance matches the provided architectures, demonstrating its effectiveness.
>
> |Architectures|CIFAR-10|CIFAR-100|ImageNet-16-120|
> |:-:|:-:|:-:|:-:|
> |40%-50%|75.81|47.13|15.66|
> |80%-90%|88.37|60.22|35.30|
> |top-20|94.36|73.51|46.34|
>
> ## Weakness
> >Dependency on prior knowledge quality
>
> The dependency is discussed in weakness 1 in Reviewer PwDk.
> >Limited architectural diversity
>
> We discuss the diversity in claims and evidence 1.
> >Lack of Dataset Conditioning
>
> Indeed, PG-NAG has performed the dataset conditioning. Specifically, we incorporate constraints to guide the process for different datasets to achieve a similar effect as dataset conditioning. Unlike DiffusionNAG, PG-NAG successfully applied to a wider range of tasks such as speech recognition in Table 4 of the main text.
> >Attribution of performance gains
>
> The performance gains of PG-NAG are mainly attributed to the guidance mechanism instead of the high-quality training data. The effectiveness of the guidance mechanism is demonstrated in claims and evidence 2.
>
> ## Other Comments
> >Diversity of generated architectures
>
> We investigate the diversity regarding the parameters and distribution of generated architectures. A visualization of the accuracy and params of 20 generated architectures is in Figure(_https://anonymous.4open.science/r/PGNAG/generated%20architectures%20in%20NAS-Bench-201.pdf_), it demonstrates that PG-NAG can generate diverse architectures.
> > Include DiffusionNAG
>
> We added the mean accuracy of DiffusionNAG on CIFAR-10 (97.39%$\pm$0.01), which is lower than PG-NAG (97.48%$\pm$0.08).
> >Potential benefits of incorporating dataset-specific knowledge
>
> Incorporating dataset-specific knowledge could enhance accuracy. PG-NAG achieves top performance on NAS-Bench-101 in Appendix Table 1. To compare, we remove architectures from NAS-Bench-101 in prior knowledge and the results can be seen below. This shows that dataset-specific knowledge could make PG-NAG more precise.
>
> |Prior knowledge|Ranking|Acc(%)|
> |:-:|:-:|:-:|
> |no NAS-Bench-101|0.004|94.08|
> |PG-NAG|0.001|94.23|
>
> ## Questions
> >Architecture retrieval
>
> * Differences: A retrieval method needs to learn the target dataset to obtain the conditions. In contrast, PG-NAG learns from existing benchmarks and then transfers the learned design principles across different tasks, without the learning process on the target dataset.
> * Ensure the novelty: PG-NAG introduces noise to explore variations beyond the benchmarks. Figure(_https://anonymous.4open.science/r/PGNAG/differences%20in%20generated%20architecture%20in%20TransNAS-bench-101%20and%20prior%20knowledge.png_) illustrates the generated architectures for TransNAS-Bench-101 are different from the learned architectures in NAS-Bench-201.
>
> >Variance and diversity
>
> Please see weakness 2.
> >Comparison with DiffussionNAG
>
> * Ensure generating high-performance architecture: We replace the predictor with a guidance mechanism, whose effectiveness has been discussed in claims and evidence 2.
> * Reason for performance gains: The performance gains cannot be attributed to the decent performance of training architectures on unseen datasets, it should be attributed to the effectiveness of the guidance mechanism based on prior knowledge. For example, NAS-Bench-NLP includes linear operations absent in prior knowledge, showing generated architectures adapt to new tasks. Additionally, prior knowledge is used for image classification, our tasks include speech recognition and natural language processing.
>
> >Incorporating data information
>
> * Challenges brought by incorporating dataset include reduces in generalization and the need for encoding. PG-NAG aims to generalize across datasets rather than overfitting to a specific one. While dataset-specific information can improve accuracy, it may reduce generalizability. Additionally, PG-NAG learns architecture design principles from different datasets, these architectures need to be represented in a unified format.
> * The difficulties preliminary come from the need to choose appropriate datasets and ensure effective learning across them. First, we need to select which dataset to be incorporated. To enhance the generalization, we choose three widely used NAS benchmarks. Second, to ensure effective learning across datasets, we encode architectures using a unified DGL graph representation.

---

### Official Review · Reviewer_PwDk · 2025-03-13

**Overall Recommendation:** 4

**Summary:**

This paper proposes a method, Prior Knowledge Guided Neural Architecture Generation, to efficiently generate high-performance neural architectures without the need for an exhaustive search and evaluation process. The key idea is to leverage prior knowledge extracted from high-performance architectures to guide a diffusion model for architecture generation. The method is validated on several search spaces, including DARTS, NATS-Bench, TransNAS-Bench-101, NAS-Bench-ASR, and NAS-Bench-NLP, achieving state-of-the-art performance with significantly reduced computational cost (0.004 GPU days for ImageNet).

## update after rebuttal

After the rebuttal, I feel my concerns have been well-discussed by the authors, and some of weaknesses has been fixed. Therefore, I choose to raise my positive scores. As we see, the majority of reviewers are lean towards to accept this submission, maybe we ask the Reviewer xYiC who holds the only negative score to further discuss if his/her concerns have been addressed.

**Claims And Evidence:**

Some claims (e.g., efficiency and accuracy) are well-supported by experimental results.

**Essential References Not Discussed:**

This paper already has a comprehensive literature review.

**Experimental Designs Or Analyses:**

Yes, I reviewed the soundness and validity of the experimental design and analysis in the paper, focusing on the benchmark selection, comparison methods, performance metrics, and ablation studies.

**Methods And Evaluation Criteria:**

Yes, the proposed methods and evaluation criteria generally make sense for the problem of neural architecture generation.

**Other Comments Or Suggestions:**

Refer to the Weaknesses section

**Other Strengths And Weaknesses:**

Other Strengths,
**Efficiency.** PG-NAG eliminates the need for costly architecture evaluations, requiring only 0.004 GPU days to generate architectures with competitive performance.
**The use of Shapley values.** Shapley values to quantify the contribution of each operation and connection is new and grounded in cooperative game theory.
**Generality.** PG-NAG shows strong performance across diverse search spaces and tasks, including vision, speech, and language.

Other Weaknesses,
**Limited Discussion on Prior Knowledge.** The manuscript does not provide enough detail on the potential limitations or biases introduced by relying on prior knowledge from existing benchmarks.
**Fixed Noise Schedule.** The noise schedule in the diffusion model is fixed (sigmoid). However, different search spaces and architecture complexities might require different noise schedules.
**Size and Performance Trade-off.** The paper focuses heavily on accuracy but does not explore the trade-off between model size and accuracy.
**Improve Interpretability.** The manuscript should Include analysis or visualization of why the generated architectures perform well.

**Questions For Authors:**

Refer to the Weaknesses section

**Relation To Broader Scientific Literature:**

The submission provides the discussion of the relevant literature. Prior work required iteratively evaluating a large number of architectures, and the manuscript provides a more efficient way to automatically generate architectures.

**Theoretical Claims:**

Yes, I examined the theoretical claims related to the use of Shapley values(Equations 3 and 4) and the diffusion model (Equations 5, 6, and 7).

---

> ### Author Rebuttal · Authors · 2025-03-31
>
> Thank you for recognizing our efficiency, novel use of Shapley values, and generality. We are also grateful for the valuable feedback.
>
> ## Weakness
>
> > Limited discussion on prior knowledge
>
> A potential limitation is that the quality of prior knowledge affects the accuracy of generated architectures. Specifically, we compare different strategies for obtaining prior knowledge. In our main text, we select each top-20 high-performance architectures from the three benchmarks. To compare with it, we use the following sample strategies: architectures uniformly sampled across a performance range of 0%–100%, architectures uniformly sampled from two narrower ranges (i.e., 90%-100% and 80%–90%). The architecture accuracy we used is the validation accuracy in CIFAR-10. To see the distribution of architectures more intuitively, we take the benchmark NAS-Bench-101 as an example and visualize the performance and parameter distribution of architectures under different strategies in the Figure (_https://anonymous.4open.science/r/PGNAG/prior.png_). As shown in the table below, architectures randomly sampled in a narrow range of high performance can also generate high-performance architecture. This demonstrates that the quality of prior knowledge is crucial for the effectiveness of PG-NAG.
>
> |Method|CIFAR-10|CIFAR-100|ImageNet-16-120|
> |:-:|:-:|:-:|:-:|
> |sample in 0%-100%|87.11|56.95|30.57|
> |sample in 80%-90%|88.37|60.22|35.30|
> |sample in 90%-100%|93.23|70.06|43.02|
> |top-20|94.36|73.51|46.34|
>
> > Fixed noise schedule
>
> The sigmoid noise schedule is fixed because it is effective when different search spaces and architecture complexities are adopted. Specifically, we conduct experiments in NAS-Bench-201 and DARTS search spaces, comparing the performance of sigmoid, linear, and cosine noise schedules. We evaluate PG-NAG on CIFAR-10, CIFAR-100, and ImageNet-16-120 in NAS-Bench-201, and evaluate PG-NAG on CIFAR-10 in DARTS. The table below shows the results that the sigmoid schedule achieves the best performance regarding different search spaces and architecture complexities.
>
> | Noise Schedule | CIFAR-10 | CIFAR-100 | ImageNet-16-120 | CIFAR-10 in DARTS |
> |:-:|:-:|:-:|:-:|:-:|
> |linear|93.42|   70.90   |      45.33      |       97.40       |
> |cosine| 93.50 |   70.67   |      44.53      |       97.54       |
> |sigmoid|  94.36   |   73.51   |      46.34      |       97.56       |
>
> > Size and performance trade-off
>
> We would like to clarify that PG-NAG can handle the trade-off between model size and accuracy. This is because the prior knowledge already covers architectures with both high performance and diverse parameter counts, as visualized in Figure (_https://anonymous.4open.science/r/PGNAG/generated%20architectures%20in%20NAS-Bench-201.pdf_). We control the model size of the generated architectures when needed. As shown in the Table below, in the first three lines we control the model size and the last one is the results we reported in the main text, From the results, we can find that when the model size becomes smaller, the performance of PG-NAG does not drop significantly, demonstrating its ability to balance model size and performance effectively.
>
> | FLOP (M) | Params (MB) | Latency (ms) | CIFAR-10 | CIFAR-100 | ImageNet-16-120 |
> |:-:|:-:|:-:|:-:|:-:|:-:|
> |121.82|0.858|21.41|93.50|70.67|44.53|
> |149.34|1.045|19.97|94.02|72.99| 45.44 |
>  |  153.27  |    1.073    |    20.22     |   94.37     |   73.22   |      46.71      |
>  |  184.73  |   1.289     |    20.59     |    94.36     |   73.51   |      46.34      |
>
> > Improve interpretability
>
> To interpret why the generated architectures perform well, we add two visualizations showing the impact of operations on architecture performance and the correlation between operations, respectively. Furthermore, we analyze that prior knowledge can effectively ensure the generation of high-performance architectures.
> * Operation-wise Influence on Performance: In Appendix Figure 7, we visualize the impact of different operations on architecture performance across various search spaces. This helps illustrate the contribution of each operation to overall accuracy.
> * Relevance of Operations in Benchmarks: We provide visualizations of operation relevance in NAS-Bench-101 (_https://anonymous.4open.science/r/PGNAG/Shapley%20values/NAS-Bench-101.pdf_), NAS-Bench-201(_https://anonymous.4open.science/r/PGNAG/Shapley%20values/NAS-Bench-201.pdf_), and NAS-Bench-301 (_https://anonymous.4open.science/r/PGNAG/Shapley%20values/NAS-Bench-301.pdf_), which are used to obtain prior knowledge. Shapley values quantify the marginal contribution of each operation, while heatmaps highlight the correlations between different operations for better interpretability.
> * Analysis of the effectiveness of PG-NAG:
> To further explain why our method performs well, we provide a detailed discussion in response to Reviewer WwRR’s Weakness 1, analyzing the interactions between architecture components and their impact on performance.

---

### Official Review · Reviewer_WwRR · 2025-03-13

**Overall Recommendation:** 4

**Summary:**

This paper proposes a neural architecture generation method called Prior Knowledge Guided Neural Architecture Generation (PG-NAG), which aims to generate high-performance neural architectures without the need for search and evaluation processes. By quantifying the contribution of each component within an architecture to its overall performance, the method identifies valuable prior knowledge and uses it to guide a diffusion model to generate architectures for various tasks. Extensive experiments demonstrate that PG-NAG achieves superior accuracy with minimal computational resources (e.g., generating architectures with 76.1% top-1 accuracy on ImageNet and 97.56% on CIFAR-10 in just 0.004 GPU days). The method also shows strong generalization across unseen search spaces like TransNAS-Bench-101 and NATSBench.

## Update after rebuttal

The authors' rebuttal looks great to me. I am finally happy to raise the recommendation to clear accept.

**Claims And Evidence:**

The claims made in the submission are supported.

**Essential References Not Discussed:**

No significant references are missing.

**Experimental Designs Or Analyses:**

The experimental setup and analyses appear to be well-structured and appropriate for assessing the claims made.

**Methods And Evaluation Criteria:**

Methods and evaluation criteria are appropriate.

**Other Comments Or Suggestions:**

Please see Strengths And Weaknesses.

**Other Strengths And Weaknesses:**

Pros:

1. The manuscript introduces a new approach to neural architecture generation by leveraging prior knowledge to guide the diffusion model, eliminating the need for traditional search and evaluation processes.
2. This work achieves high performance with extremely low computational costs (0.004 GPU days).
3. Extensive experiments and comparisons with baseline, thoroughly validating the effectiveness and efficiency of PG-NAG.

Cons:

1. Providing deeper analysis and insights into why the proposed method works effectively can enhance the quality of the manuscript.
2. The manuscript involves evaluations across multiple tasks and benchmarks, which strengthens the robustness of the work.  However, this also introduces a multitude of metrics.  A detailed explanation of these metrics would be beneficial for understanding the contributions of the manuscript.
3. The prior knowledge extracted from existing benchmark datasets may not always represent the target tasks or domains. Further exploration is needed to understand how the selection of benchmark datasets impacts the effectiveness of the generated architectures
4. The motivation behind using Shapley values to quantify the contributions of components within an architecture needs to be better articulated.
5. Highlighting the best-performing results would make it easier for readers to quickly identify the strengths of their method.

**Questions For Authors:**

1. How can the quality and diversity of prior knowledge be ensured when selecting high-performance architectures from benchmark datasets?
2. How does PG-NAG handle the trade-off between computational efficiency and the accuracy of generated architectures, especially when increasing the number of diffusion steps?

**Relation To Broader Scientific Literature:**

The paper is well-aligned with recent literature, like neural architecture search and diffusion models.

**Theoretical Claims:**

There are no explicit theoretical proofs or claims that require verification. The focus of the manuscript is primarily on the empirical evaluation of PG-NAG rather than on theoretical analysis.

---

> ### Author Rebuttal · Authors · 2025-03-31
>
> Thank you for recognizing our good performance, effectiveness, and efficiency. We are also grateful for the valuable and constructive feedback.
> ## Weaknesses
> >An analysis of how the method works effectively
>
> The effectiveness of PG-NAG can be attributed to prior knowledge guidance, operation feature extraction, and connection feature extraction. As shown in Table 6 in the main text, skipping or replacing any of these results in a decline in performance. Specifically, prior knowledge measures the contribution of each operation or connection in an architecture with Shapley values. Moreover, the operation feature extraction effectively learns representations of operations that contribute significantly to performance. These learned features are then integrated into the generation process. Similarly, the connection feature extraction learns the connections that are useful to high-performance architectures, ensuring the generation of high-performance architectures.
> >An explanation of metrics used in PG-NAG
>
> The explanations of the six metrics used are in Table (_https://anonymous.4open.science/r/PGNAG/metrics.md_). We added this table in the new manuscript.
> >How does the selection of benchmarks impact the effectiveness of generated architectures
>
> We leverage NAS-Bench-101, NAS-Bench-201, and NAS-Bench-301 to ensure the generated architectures perform well in basic image processing and large-scale complex tasks, exhibiting strong generalization capabilities across other tasks.
> * NAS-Bench-101 focuses on image classification, ensuring the generated architectures perform well on basic image tasks. The experiment shows that prior knowledge without NAS-Bench-101 has lower performance (comments 3 of VjZy).
> * NAS-Bench-201 contains simple but effective operations and has been the basis for various search spaces. Therefore, it can guarantee the generated architectures have generalization capabilities across different tasks.
> * NAS-Bench-301 supports complex architecture design for large-scale tasks, ensuring that the generated architectures are suitable for complex tasks like ImageNet.
>
> >The motivation for Shapley values
>
> The motivation behind using Shapley values lies in the cooperative relationship between operations and connections in architecture. Specifically, the operations and connections in architecture are not independent of each other, and they interact as a whole to determine the overall performance. Shapley value can quantify the contribution in a cooperative game, thus helping us identify the contribution of operations and connections to the architecture performance [1]. By incorporating this, PG-NAG can focus on generating architectures that emphasize these critical components, leading to higher-performance designs.
>
> >Highlight the best-performing results
>
> We have highlighted the best results in the new manuscript.
> ## Questions
> >How to ensure the quality and diversity of prior knowledge?
>
> The quality and diversity of prior knowledge are ensured by the benchmark selection, diversity, and sampling method.
> * Selection: We select high-performance architectures from three widely used NAS benchmarks, including NAS-Bench-101, NAS-Bench-201, and NAS-Bench-301. These architectures achieve good performance on well-known image classification datasets such as ImageNet. This allows us to learn how components are comprised in high-performance architectures and apply this knowledge to other search spaces.
> * Diversity: To illustrate the diversity of selected benchmarks, Figure (_https://anonymous.4open.science/r/PGNAG/prior.png_) visualizes the distribution of $20$ architectures sampled from NAS-Bench-101, confirming that our selected architectures cover a diverse distribution in terms of both accuracy and complexity.
> * Sampling method: We conduct ablation studies on the number of selected architectures in Table 8 and the selection method (weakness 1 in Reviewer PwDK). The results confirm that PG-NAG utilizes an effective sampling method to generate high-performance architectures.
>
> >How does PG-NAG balance computational efficiency and accuracy?
>
> PG-NAG addresses this trade-off by using a sigmoid noise schedule and prior knowledge to reduce diffusion steps.
> Firstly, we conduct experiments on CIFAR-10 in NAS-Bench-201 about how diffusion steps impact performance. Results below show that beyond $1,000$ steps, performance gains are minimal compared to the added computational cost.
>
> |time steps|CIFAR-10|
> |:-:|:-:|
> |$800$|$94.22$|
> |$1,000$|$94.36$|
> |$1,200$|$94.37$|
>
> Secondly, to achieve high performance in fewer steps, we use a sigmoid noise schedule for a smoother and more controlled noise removal process. Additionally, by integrating prior knowledge, PG-NAG effectively reduces the search space and enables the diffusion model to focus on high-performance designs without added computational cost.
>
> **Reference**
>
> [1] Shapley-NAS: Discovering operation contribution for neural architecture search, CVPR'22

---

> > ### Comment · Reviewer_WwRR · 2025-04-05
> >
> > The authors have addressed my concerns well, I acknowledge that the authors' contributions on the efficiency,  performance, and their efforts on quality and diversity. Therefore I have increase my score to 4.

---

> > > ### Author Response · Authors · 2025-04-05
> > >
> > > Thank you for your valuable comments and positive feedback.
> > > Your insightful comments have substantially enhanced the clarity and overall quality of our paper. We appreciate your time and effort in reviewing our work. Thanks!

---

### Decision · Program_Chairs · 2025-05-01

**Decision:**

Accept (poster)

**Comment:**

Three reviewers recommend acceptance (2 Accept and 1 Weak Accept) and one rejection (Weak Reject). The reviewers acknowledge the effectiveness, efficiency, and generality of the proposed method, and the extensive experiments. However, they raised some concerns about the justification of the method and the chosen metrics, the limited discussion of potential limitations, the fairness of the comparisons in Table 1. This latter point was raised by Reviewer xYiC, but the authors further clarified it in a private answer to the AC because they were at that point unable to respond to the reviewer directly. Ultimately, despite recommending a Weak Reject, Reviewer xYiC acknowledges that this paper could be accepted. The balance leans even further towards acceptance considering the authors' explanations of the results in Table 1 in the private comment to AC. As such, the AC recommends acceptance but strongly encourages the authors to include elements of their feedback, including the clarification of the comparison fairness, in the final version.